# Diet, Physical Activity, and Disinhibition in Middle-Aged and Older Adults: A UK Biobank Study

**DOI:** 10.3390/nu13051607

**Published:** 2021-05-11

**Authors:** Lizanne J. S. Schweren, Daan van Rooij, Huiqing Shi, Henrik Larsson, Alejandro Arias-Vasquez, Lin Li, Liv Grimstvedt Kvalvik, Jan Haavik, Jan Buitelaar, Catharina Hartman

**Affiliations:** 1Interdisciplinary Center Psychopathology and Emotion Regulation, University Medical Center Groningen, 9700 RB Groningen, The Netherlands; c.hartman@accare.nl; 2Donders Center for Brain, Cognition and Behaviour, Department of Cognitive Neuroscience, RadboudUMC, 6525 GA Nijmegen, The Netherlands; d.vanrooij@donders.ru.nl (D.v.R.); huiqing.shi@radboudumc.nl (H.S.); Alejandro.AriasVasquez@radboudumc.nl (A.A.-V.); jan.buitelaar@radboudumc.nl (J.B.); 3School of Medical Sciences, Örebro University, S-701 82 Örebro, Sweden; Henrik.Larsson@ki.se (H.L.); lin.li@oru.se (L.L.); 4Department of Medical Epidemiology and Biostatistics, Karolinska Institutet, SE-171 77 Stockholm, Sweden; 5Department of Biomedicine, Public Health and Primary Care, University of Bergen, NO-5020 Bergen, Norway; liv.kvalvik@uib.no (L.G.K.); Jan.Haavik@uib.no (J.H.); 6Bergen Centre of Brain Plasticity, Haukeland University Hospital, NO-5012 Bergen, Norway; 7Karakter Child and Adolescent Psychiatry University Centre, RadboudUMC, 6525 GA Nijmegen, The Netherlands

**Keywords:** dietary habits, behavioral disinhibition, brain health, physical activity, prudent diet

## Abstract

Disinhibition is a prominent feature of multiple psychiatric disorders, and has been associated with poor long-term somatic outcomes. Modifiable lifestyle factors including diet and moderate-to-vigorous physical activity (MVPA) may be associated with disinhibition, but their contributions have not previously been quantified among middle-aged/older adults. Here, among N = 157,354 UK Biobank participants aged 40–69, we extracted a single disinhibition principal component and four dietary components (prudent diet, elimination of wheat/dairy/eggs, meat consumption, full-cream dairy consumption). In addition, latent profile analysis assigned participants to one of five empirical dietary groups: prudent-moderate, unhealthy, restricted, meat-avoiding, low-fat dairy. Disinhibition was regressed on the four dietary components, the dietary grouping variable, and self-reported MVPA. In men and women, disinhibition was negatively associated with prudent diet, and positively associated with wheat/dairy/eggs elimination. In men, disinhibition was also associated with consumption of meat and full-cream dairy products. Comparing groups, disinhibition was lower in the prudent-moderate diet (reference) group compared to all other groups. Absolute βs ranged from 0.02–0.13, indicating very weak effects. Disinhibition was not associated with MVPA. In conclusion, disinhibition is associated with multiple features of diet among middle-aged/older adults. Our findings foster specific hypotheses (e.g., early malnutrition, elevated immune-response) to be tested in alternative study designs.

## 1. Introduction

Behavioral disinhibition is the tendency to act in an uncontrolled fashion, without prior risk-assessment and/or in disregard of social conventions. Manifestations of disinhibition may include poor self-regulation, impulsivity, compulsivity, emotional instability, aggression, and high-risk behaviors [1]. Extreme manifestations of disinhibition are at the core of severe and often persistent psychiatric disorders, including attention-deficit/hyperactivity disorder (ADHD), mania, and addiction. Poor self-regulation and psychiatric disorders related to disinhibition have been associated with a range of negative long-term health outcomes such as obesity [2] and cardiovascular disease [3], emphasizing the relevance of the concept.

Several lines of research indicate that nutrition may be an important factor in the association between (extreme manifestations of) disinhibition and poor health. Impulsivity and poor self-regulation have been associated with unhealthy food choices, both in experimental settings [4,5] and in observational studies [6]. Similarly, overall diet quality is poorer in patients with ADHD [7], mania [8], and addictions [9] compared to unaffected individuals. Disinhibited eating, or the tendency to overeat in response to different stimuli including external food cues and stress, is associated with higher energy intake, higher BMI, and elevated obesity risk [10,11], as well as with more general indicators of behavioral disinhibition such as ADHD symptoms [12,13] and poor executive functioning task performance [14,15].

While a causal pathway running from disinhibition/self-regulation to dietary habits is plausible, nutrition may also contribute to disinhibited behaviors. Rodents being fed a high-sugar or high-fat diet for a prolonged period of time showed increases in both impulsive [16] and compulsive behaviors [17]. Excessive consumption of palatable foods may evoke lasting changes in cortico-limbic dopamine signaling that are reminiscent of patterns associated with addiction [18]. Dopaminergic cortico-limbic systems are implicated not only in addiction, but also in emotion regulation, reward processing, and impulse control. Others have speculated that disinhibited behaviors such as those seen in ADHD might arise as a result of diet-induced inflammation [19] or hypersensitivity to specific foods/allergens [20]. 

The complex relationship between nutrition and disinhibition likely extends to other lifestyle domains, including physical activity. Regular physical exercise affects levels of cortisol, oxidative stress, and inflammation, each of which may influence cognitive functioning (e.g., [21]). Lower levels of moderate-to-vigorous physical activity (MVPA) have been associated with poor self-control/disinhibition [22,23,24], and long-term physical activity interventions have been proposed to ameliorate inhibitory control task performance and reduce symptoms of psychiatric outcomes related to disinhibition (e.g., [25,26]).

In the current study, we investigated associations between disinhibition and dietary behaviors in the unprecedentedly large UK Biobank sample. The observational and cross-sectional nature of our study does not allow causal conclusions regarding the relationship between diet and disinhibition, nor can we test mediation effects (e.g., from disinhibition to diet to poor health). By addressing several important limitations of the existing epidemiological evidence, however, we affirm and broaden the knowledge base of the potentially causal bidirectional relationships between diet and disinhibition. First, prior studies have primarily focused on overall diet quality; associations between disinhibition and other dietary features remain largely unexplored. Here, we investigated associations between disinhibition and multiple data-driven dietary patterns, and compared homogenous dietary subgroups. Second, behavioral disinhibition is often studied in the young, especially in boys and young men. Dietary patterns differ by sex and change with age [27,28], as may manifestations of disinhibition. We studied disinhibition among middle-aged and older adults, allowing investigation of its lifestyle correlates at different ages and for men and women separately. Third, dietary habits inevitably cluster together with other health behaviors, including physical activity [29]. Prior epidemiological studies, however, focused on either nutrition or physical activity levels. Here, combined modelling of dietary habits and physical activity allows quantification of their unique and shared contributions.

## 2. Materials and Methods

### 2.1. The UK Biobank Study

The data underlying this article were provided by UK Biobank. UK Biobank is a prospective cohort study providing detailed characterization of over half a million UK-based persons aged 40–69 years at recruitment (2006–2010; http://www.ukbiobank.ac.uk/wp-content/uploads/2011/11/UK-Biobank-Protocol.pdf (accessed on 7 January 2021)). With linkage to routinely collected data such as those produced by the National Health Service, UK Biobank offers a highly efficient resource for observational epidemiology [30]. UK Biobank has ethical approval from the North West Multicenter Research Ethics Committee. All participants gave informed consent. For the current study, we included participants who completed the follow-up online mental health questionnaire (MHQ) that was sent to 339,092 participants who had agreed to being contacted by email. Of those, 46% (N = 157,354) completed the assessment [31].

### 2.2. Diet

Dietary habits were assessed using a 29-item food frequency questionnaire (FFQ) about participants’ past-year average diets. While this relatively short FFQ does not allow calculation of total energy or (macro-)nutrient intake, it has been shown to reliably estimate intakes of the main food groups [32]. Individual FFQ items are highly correlated reflecting underlying dietary patterns [33]. We thus performed two established data-reduction techniques: principal component analysis (PCA) to detect shared variance between items (i.e., dietary components), and latent profile analysis (LPA) to detect clusters of subjects behaving similarly with respect to the dietary components (i.e., dietary groups). Both techniques are complementary: while PCA extracts common features of diet across participants (e.g., “overall diet quality”), LPA extracts homogenous participant groups best described by (combinations of) these features. For details, see Appendix A.

Raw data included continuous variables (e.g., tablespoons of vegetables), ordinal variables (e.g., cereal intake frequency), food type descriptions (e.g., bread type: brown/white/whole-meal/other), and one tick-box item assessing elimination of specific food groups (eggs/dairy/wheat/sugar). Alcohol-related items were removed, since heavy drinking contributed to our outcome of interest (disinhibition). Binary contrasts were created for each food type (e.g., ‘brown bread’ vs. ‘any other bread type’) and elimination item (e.g., ‘I never eat eggs’ vs. ‘no restrictions’). The resulting 58 food items/contrasts, regressed on age and sex, were entered in a PCA with Promax rotation on the residuals. Component selection was performed manually (LJSS), starting with a single component and adding components one-by-one. Components were retained as long as they were unique (no prior component reflected the same set of items), interpretable (items loading on the component form plausible groupings), and stable (the component remained upon adding the next component). Component selection was verified with a second author (CAH).

Next, we performed latent profile analysis based on the dietary component scores to derive relatively homogenous participant groups. R-package ‘mclust’ [34] performs model-based clustering of participants based on parameterized finite Gaussian mixture models. We defined no a priori model constraints, and allowed a maximum of 16 clusters or subject groupings. We performed 100 permutations and calculated Light’s generalisation of Cohen’s kappa’s as an indicator of acceptable (kappa > 0.8) model stability. Model fit was assessed using Bayesian information criterion (BIC), striving for maximal parsimony without compromising model fit. In addition, each cluster should represent no less than 1% of the full sample, and the model solution should be both interpretable and plausible. Once the optimal model had been identified, group membership of each participant was set to the mode of classifications across 1000 permuted model fittings. 

### 2.3. Behavioral Disinhibition

To assess disinhibition as a unitary construct, we performed a PCA on all disinhibition-related items. For details, see Appendix A. First, all data-fields related to disinhibition, impulsivity, compulsivity, and/or emotional instability were selected. Selected data-fields covered addictions including smoking, risk behaviors such as heavy drinking, self-reported and hospital-record diagnoses of selected psychiatric disorders, self-harm behaviors and personality questionnaire items. To ascertain a balanced representation of different manifestations of disinhibition, we defined nine categories of disinhibited behaviors: smoking, addiction (other than smoking), excessive cannabis use, personality factors, self-harm, mania, obsessive compulsive behaviors, externalizing behaviors, and risk-taking behaviors including binge-drinking. Next, we performed a PCA based on tetrachoric correlations between these behaviors. The single-component model, preferred a priori, presented with no interpretational shortcomings: all behaviors loaded positively on the principal component with factor loadings ranging from 0.335 to 0.708 (Appendix A). For each subject, a factor score was extracted with higher scores indicating a higher tendency for disinhibition.

### 2.4. Physical Activity

Participants self-reported their frequency (days/week) and duration (minutes/day) of engaging in moderate and vigorous physical activity. Questions were adapted from the IPAQ, an internationally validated instrument [35]. In the questionnaire, moderate physical activity (MPA) was described as “activities like carrying light loads, cycling at normal pace (do not include walking) for 10 min or more.” Vigorous physical activity (VPA) was described as “activities that make you sweat or breathe hard such as fast cycling, aerobics, heavy lifting for 10 min or more.” MVPA was calculated as the sum of moderate and vigorous physical activity in minutes/week. MVPA was strongly skewed, ranging from 0 to 8640 min/week with a median of 180 min/week and 80% of participants being active for ≤480 min/week. MVPA was thus converted to pseudo-continuous quintile scores: 1: 0–43 min/week; 2: 43–120 min/week; 3: 120–240 min/week; 4: 240–480 min/week; 5: >480 min/week. 

In sensitivity analyses, we also evaluated moderate and vigorous physical activity separately. For MPA, quintile scores were created as before (1: 0–30 min/week, 2: 30–80 min/week, 3: 80–151 min/week, 4: 151–360 min/week, 5: >360 min/week). A large proportion of the sample never performed VPA; thus, VPA was split in three groups (1: never; 2: 0-median i.e., 0–80 min/week; 3: >80 min/week).

### 2.5. Covariates

Participants self-reported their ethnicity as white, mixed, Asian, Black, Chinese, or other, which was recoded to white/non-white. Body mass index (BMI) in kg/m^2^ was measured on site. Participants self-reported their habitual sleep duration in hours and minutes. As both long and short sleep duration are believed to be detrimental, sleep duration was categorized in three non-ordinal categories. Short: <7 h/night; normal: 7–9 h/night or 7–8 h/night for adults age ≥65; long: >9 h/night or >8 h/night for adults age ≥65. Neighborhood-based index of multiple deprivation (IMD) by region (England, Scotland, Wales), an index of relative poverty of small areas based on e.g., unemployment and crime rates, was determined based on postcode of residence. Participants also self-reported their annual household income, which was adjusted for household size using equivalence factors, their unemployment status, and the number of years of education [36] (no qualifications: 7y; CSEs: 10y; O-levels/GCSEs: 10y; A levels/AS levels: 13y; other professional qualification: 15y; NVQ or HNC: 19y; college or university degree: 20y). 

### 2.6. Statistical Procedures

Missing values for all variables were imputed with multivariate imputation using chained equations (R-package ‘MICE’ [37]). Subsequent steps were performed in men and women separately. First, all variables were standardized. For descriptive purposes, we next predicted disinhibition from all covariates. Next, in six single-predictor models, we predicted disinhibition from the dietary components, the dietary grouping variable, and MVPA. Initially, an age-by-predictor interaction term was included in each model as well; however, no relevant interactions were detected (Appendix A), hence age-interaction terms were discarded. To assess unique associations between disinhibition and each predictor, we ran two multiple-predictor models: one predicting disinhibition from the dietary components and MVPA, and one predicting disinhibition from the dietary grouping variable and MVPA. Again, age-interactions were discarded upon finding no relevant interaction effects. 

Alpha was conservatively adjusted for testing four DCs (see results), one dietary grouping variable with five levels (see results), and MVPA, in men and women separately: α = 0.05/((4 + (5 − 1) + 1) * 2) = 0.0028. Provided power = 0.8, α = 0.0028 and *n* ≥ 66,419, we were able to detect associations of β ≥ 0.02. Differentiation between statistically and clinically significant findings is not achieved by further lowering of α, but by investigating β. Absolute βs of 0.2 are generally referred to as ‘weak’; however, in models predicting complex behaviors, large βs are not expected and small βs can be informative [38]. Consensus regarding denotation is lacking. Here, for interpretational purposes, we qualify absolute βs of significant associations (*p* < 0.0028) between 0.02–0.19 as very weak, 0.2–0.39 as weak, 0.4–0.59 as moderate, 0.6–0.79 as strong, and ≥0.8 as very strong. Absolute β < 0.02 is qualified as not associated.

## 3. Results

### 3.1. Dietary Components

Four dietary components were extracted from the FFQ data (Table 1). DC1 reflects a prudent diet, with positive loadings for fruits, vegetables, wholegrain bread and fish, and negative loadings for refined carbohydrate products, processed meat and instant coffee. DC2 reflects elimination of wheat, dairy and/or eggs. DC3 reflects meat and to a lesser extent fish consumption, and DC4 reflects consumption of full-cream milk and butter/spreads. Correlations between components were modest (r ≤ 0.124, ST1.3).

### 3.2. Dietary Groups

The only solution with acceptable model fit and stability contained five mutually exclusive latent groups (Table 2, Appendix A). The largest group (*n* = 49,463, 31.4%) had relatively high prudent diet scores and moderate scores across the other three DCs and was set as the reference group (‘prudent-moderate’). The second group was characterized by a generally unhealthy (i.e., non-prudent) diet and high consumption of full-cream dairy products (‘unhealthy’, *n* = 42,663, 27.1%). The third group comprised individuals who avoided meat and adhered to a prudent diet (‘avoid meat’, *n* = 21,797, 13.9%). The remaining two groups were each driven by a single DC: one group comprised participants who avoided full-cream dairy products (‘low-fat dairy’: *n* = 32,555, 20.7%), and one group comprised participants who eliminated wheat/dairy/eggs but not sugar from their diet (‘restricted’; *n* = 10,876, 6.9%).

### 3.3. Disinhibition

Disinhibition scores were higher in men (M = 0.01, SD = 1.09) compared to in women (M = −0.11, SD = 1.05; β_MALE_ = 0.118, se = 0.005, *p* < 0.0001). In the covariate-only model, disinhibition was associated with younger age (men: β = −0.226, se = 0.004; women: β = −0.212, se = 0.003), unemployment (men: β = 0.215, se = 0.027; women: β = 0.275, se = 0.035), white ethnicity (men: β_NON-WHITE_ = −0.189, se = 0.022; women: β_NON-WHITE_ = −0.123, se = 0.019), long sleep duration (men: β = 0.201, se = 0.024; women: β = 0.215, se = 0.023), short sleep duration (men: β = 0.129, se = 0.009; women: β = 0.122, se = 0.008), neighborhood deprivation (men: β = 0.112, se = 0.004; women: β = 0.111, se = 0.004), and BMI (men: β = 0.040, se = 0.004; women: β = 0.036, se = 0.003). In women, disinhibition was also associated with lower adjusted income (β = −0.036, se = 0.003) and more years of education (β = 0.022, se = 0.003). 

### 3.4. Diet and Disinhibition

In the single-predictor models including all covariates, prudent diet (DC1) was associated with lower disinhibition scores in men (β = −0.036) and women (β = −0.043; Table 3). By contrast, elimination of wheat, dairy and/or eggs (DC2) was associated with higher disinhibition scores in both groups (β_MEN_ = 0.030, β_WOMEN_ = 0.038). Meat consumption (DC3) and full-cream dairy consumption (DC4) were associated with more disinhibition in men (β_DC3_ = 0.041; β_DC4_ = 0.023) but not in women. Note that no association exceeded β = 0.2, indicating very weak effects. Associations between disinhibition and each DC remained virtually unchanged in the multiple-predictor model that included all dietary components simultaneously as well as MVPA (Table 3), suggesting (a) minimal overlap between the four dietary components, and (b) that associations between disinhibition and diet were not accounted for by physical activity. 

Demographic/socioeconomic differences between the five dietary groups are shown in Table 2. Taking into account all covariates, disinhibition was significantly lower in the prudent-moderate diet group compared to all other diet groups (Table 3). In men, the difference in disinhibition compared to the prudent-moderate diet group was strongest in the restricted group (β = 0.125), weaker in the unhealthy (β = 0.087), and meat-avoiding (β = 0.075) groups, and least pronounced in the low-fat dairy group (β = 0.042). In women, relatively strong effects were found in the restricted diet group (β = 0.156) and the meat-avoiding group (β = 0.154), followed by the unhealthy group (β = 0.097) and the low-fat dairy group (β = 0.054). Again, however, all effects were qualified as very weak (β < 0.2). Model estimates were minimally affected by adding MVPA to the model (Table 3). 

### 3.5. MVPA and Disinhibition

MVPA was not associated with disinhibition in men (β = −0.007) or women (β = −0.009; Table 3). Estimates were further attenuated after adding DC1-4 or the dietary grouping variable to the model. In sensitivity analyses, the models were rerun for moderate (MPA) and vigorous (VPA) physical activity separately. In the single-predictor models, MPA was not associated with disinhibition in men (β ≤ −0.001; SE = 0.004, *p* = 0.9186) or women (β = −0.004; SE = 0.003, *p* = 0.1647), while VPA was very weakly associated with lower disinhibition in both groups (men: β = −0.023; SE = 0.004, *p* < 0.0001; women: β = −0.023; SE = 0.003, *p* < 0.0001). Adding the dietary grouping variable to the model, the association between VPA and disinhibition remained significant (men: β = −0.021; SE = 0.004, *p* < 0.0001; women: β = −0.021; SE = 0.003, *p* < 0.0001), while the association fell short of significance after adding DC1-4 (men: β = −0.018; SE = 0.004, *p* < 0.0001; women: β = −0.016; SE = 0.003, *p* < 0.0001).

## 4. Discussion

In an unprecedented sample of over 150,000 individuals aged 40 and older, we investigated the relationships between dietary habits and disinhibition. Among men and women and across the age range, adherence to a prudent diet was associated with lower disinhibition scores, while elimination of wheat, dairy, and/or eggs was associated with higher disinhibition scores. In men, disinhibition was also associated with higher habitual consumption of meat and high-fat dairy products. Classifying participants based on their multivariate diet patterns, disinhibition scores were higher in the unhealthy diet, restricted diet, meat-avoiding, and low-fat dairy groups compared to the prudent-moderate diet reference group. All associations between diet and disinhibition were very weak (β < 0.2). Physical activity was not associated with disinhibition. 

Our finding of a negative association between prudent diet and disinhibition among middle-aged and older men and women corroborates and extends the existing disinhibition literature that tends to focus on the young, especially young males. In line with findings among younger populations [6], we show that behavioral disinhibition is associated with unhealthier food choices among middle-aged and older individuals. The negative association between prudent diet and behavioral disinhibition might indicate an effect of disinhibition on diet quality (e.g., through disinhibited eating), an effect of diet quality on disinhibited behaviors (e.g., through cortico-limbic changes), or both, or neither (e.g., when the association results from confounding). We also add to the available knowledge by qualifying the association between prudent diet and disinhibition as ‘very weak’. Short sleep duration, for instance, was >2.5 times more strongly associated with disinhibition, and age and unemployment were >5 times more strongly associated. The weak association strength of this and all other associations detected in our study might be related to the relatively old age of participants. Other factors specifically relevant in older participants (e.g., frailty or health problems) might explain a relatively large proportion of variance in dietary behaviors, leaving less variability to be explained by behavioral disinhibition. The weak association strength of all findings should be kept in mind when reading the below discussion. 

Unexpectedly, the strongest lifestyle predictor of disinhibition was a diet restricted in wheat, dairy, and/or eggs. Within the restricted diet group, most individuals eliminated a single food group and each food group was equally often eliminated (wheat: 30.5%; dairy: 27.4%; eggs: 27.6%), suggesting that disinhibition is likely not associated with a lack of nutrients specifically provided by either product. Note that the restricted diet group was also characterized by lower overall diet quality compared to the prudent-moderate diet group, potentially suggesting a more general lack of nutrients or residual confounding. Participants in the restricted diet group were more often born outside the UK and of lower socioeconomic status, potentially pointing towards early life exposure to malnutrition. In animals, malnutrition early in life is known to cause impulsiveness in adulthood [39]. Alternatively, elimination of specific foods may suggest a higher prevalence of (allergic) food intolerances among disinhibited or impulsive individuals [19]. Common genetic variants have been identified between allergic diseases and psychiatric disorders including ADHD, OCD, and addiction [40]. Studies of dieting/weight control provide yet another explanation: highly-impulsive individuals tend to score higher on dietary restraint. Restraint is a cognitive concept characterized by the (successful or unsuccessful) intention to limit food intake, that has been associated with dieting, past attempts at dieting, and weight fluctuations [41]. Although participants were instructed to report dietary habits, not intentions, it is plausible that dieters might report the dietary pattern they (successfully or unsuccessfully) intend to adhere to. Finally, higher endorsement of elimination items (“Which of the following do you never eat?”) may reflect a higher tendency among high-impulsive individuals compared to low-impulsive individuals to endorse these items, even if they do not fully meet the never-criterion. 

Our findings regarding meat consumption were contradictory. In prior studies, meat consumption has been associated with emotional instability and impulsive/thrill-seeking traits [42,43], but the opposite [6,44] and null findings [45,46] have also been reported. Here, in men only, meat consumption was associated with higher disinhibition scores. In contrast, both men and women in the meat-avoiding group had higher disinhibition scores compared to those in the prudent-moderate diet group. As meat is rich in tryptophan, a precursor of dopamine, meat consumption might affect neurotransmitter levels in brain systems important for inhibitory control. Higher disinhibition in the meat-avoiding group cannot be not explained by lower intake of omega-3 fatty acids (essential fish oil nutrients with potential limited efficacy in treating symptoms of ADHD [47]): oily fish consumption in the prudent-moderate and the meat-avoiding groups were similar (data not shown). Compared to the prudent-moderate diet group, the meat-avoiding group also had higher levels of wheat/dairy/eggs restrictions, and, similar to the dietary restrictions group, a relatively large proportion of individuals of non-white ethnicity and born outside the UK. The early-malnutrition [39] and food-intolerance hypotheses [20] may thus apply to the meat-avoiding group as well. Alternatively, the decision to (partially) eliminate meat from one’s diet might be influenced by fluctuating societal trends, to which impulsive individuals may be more sensitive [48]. 

Finally, the absence of an association between MVPA and disinhibition is of note. In line with cardiovascular health benefits being greater for vigorous compared to moderate intensity activities [49], we found a very weak association between disinhibition and VPA but not MPA. Regular physical exercise might lower baseline cortisol levels [50], which in turn may be associated with lower impulsivity [51]. In addition, regular exercise is associated with lower oxidative stress and a pro-inflammatory state that can contribute to better cognitive task performance [21]. However, the association between VPA and disinhibition was partially accounted for by dietary factors. Leisure time MVPA may be associated with better outcomes compared to occupational MVPA ([52]), but our data did not allow stratification by MVPA-type. The absence of associations with MVPA might be related to the relatively old age of the current sample, as health problems and frailty might restrict their activities. Thorough investigation of specific physical activity characteristics is beyond the scope of the current paper, and is recommended for future studies. 

Strengths of the current study include the application of state-of-the-art data-reduction techniques across modalities to capture complex latent variables. The so-created disinhibition scale robustly showed the expected associations with sociodemographic variables. Associations between diet and disinhibition were assessed both at the food-group level and at the level of multivariate dietary patterns, ensuring ecological validity, and were adjusted for other lifestyle parameters. Finally, socioeconomic status was modelled at the societal-, household- and individual level, lowering the risk of residual confounding. The current study has limitations as well. The cross-sectional and observational nature of our study precludes causal inferences. Observed associations might reflect an effect of disinhibition on dietary choices, an effect of dietary intake on disinhibited behaviors, or both. In future studies, integrating structural brain imaging data could be an important asset to elucidate the mechanisms underlying the observed associations. Another inherent vulnerability of observational studies is the possibility of unmeasured confounding, i.e., both disinhibition and dietary behaviors might be driven by an unmeasured third variable (e.g., genetic confounding [53]). We were unable to adjust analyses for total energy intake, as the short FFQ does not allow calculation of this metric. Moreover, no data is available regarding nutritional history or exposure to early life malnutrition, both of which may have an impact on normal brain development. Finally, dietary habits and physical activity levels were self-reported and might be subject to systematic bias [54] (e.g., less accurate reporting in high-impulsive participants).

## 5. Conclusions

We found significant and age-independent associations between dietary habits and disinhibition among middle-aged and older men and women. Although all associations were of very weak strength, our findings generate mechanistic hypotheses that might be tested in alternative study designs.

## Figures and Tables

**Table 1 nutrients-13-01607-t001:** Factor loadings (≤−0.3 or ≥0.3) per dietary component (DC1-4). For the full table, see ST1.2.

	DC1	DC2	DC3	DC4
Wholegrain bread vs. any other bread	0.58			
Dried fruit frequency	0.48			
Oily fish frequency	0.47			
Raw vegetables/salad frequency	0.45			
Fresh fruit frequency	0.45			
Cooked vegetables frequency	0.38			
Water frequency	0.34			
Non-oily fish frequency	0.33		0.33	
Cereal frequency	0.31			
Ground coffee vs. any other coffee	0.32			
Instant coffee vs. any other coffee	−0.31			
Refined sugar-sweetened cereal products	−0.33			
White bread vs. any other bread	−0.55			
‘I never eat dairy’ vs. no restrictions		0.64		
‘I never eat wheat’ vs. no restrictions		0.64		
‘I never eat eggs, sugar, wheat or dairy’ vs. no restrictions		0.51		
‘I never eat eggs’ vs. no restrictions		0.50		
‘I never eat dairy’ vs. no dairy restrictions		0.34		
‘I never eat wheat’ vs. no wheat restrictions		0.34		
‘I never eat sugar’ vs. no restrictions		−0.31		
Bread vs. ‘I never eat bread’		−0.38		
Lamb frequency			0.75	
Beef frequency			0.73	
Pork frequency			0.73	
Processed meat frequency			0.48	
Poultry frequency			0.40	
Added salt frequency			0.31	
Fat content of milk				0.83
Semi-skimmed milk vs. any other milk				0.55
Full cream milk vs. any other milk				0.31
Skimmed milk vs. any other milk				−0.84

**Table 2 nutrients-13-01607-t002:** Descriptive statistics per dietary group.

	M (SD)/N(%)	M(SD)/N(%)	M(SD)/N(%)	M(SD)/N(%)	M(SD)/N(%)
	Prudent-Moderate	Unhealthy	Low-Fat Dairy	Avoid Meat	Restricted
N	49,463 (31.4)	42,663 (27.1)	32,555 (20.7)	21,797 (13.9)	10,876 (6.9)
Male	20,315 (41.1)	20,143 (47.2) ^a^	12,054 (37.0) ^a^	11,287 (51.8) ^a^	4461 (41.0) ^a^
Age	55.5 (7.7)	55.9 (7.8) ^bc^	56.3 (7.6) ^bc^	56.2 (7.9) ^ab^	56.4(7.7) ^bc^
BMI	26.2 (4.3)	27.4 (4.7) ^bc^	27.3 (4.7) ^bc^	26.1 (4.3) ^c^	26.7 (4.8) ^bc^
IMD	13.4 (10.7)	15.7 (12.6) ^bc^	14.5 (11.7) ^bc^	15.5 (12.1) ^bc^	16.6 (13.2) ^bc^
Years in education	17.1 (4.1)	15.6 (4.7) ^bc^	16.3 (4.5) ^bc^	17.0 (4.3) ^bc^	16.0 (4.7) ^bc^
Annual income ^d^	32.9 (16.0)	29.1 (14.9) ^bc^	31.8 (16.0) ^bc^	32.2 (16.7) ^b^	30.6 (16.4) ^bc^
Unemployment	524 (1.1)	672 (1.6) ^b^	394 (1.2)	289 (1.3)	177 (1.6) ^b^
Non-white ethnicity	956 (1.9)	1136 (2.7) ^bc^	697 (2.1) ^b^	1209 (5.6) ^bc^	561 (5.2) ^bc^
Sleep duration					
Short	9592 (19.4)	9762 (22.9) ^bc^	7271 (22.3) ^bc^	4862 (22.3) ^bc^	2831 (26.0) ^bc^
Long	782 (1.6)	1111 (2.6) ^bc^	721 (2.2) ^bc^	527 (2.4) ^bc^	326 (3.0) ^bc^
DC1	0.55 (0.62)	−0.92 (0.72)	0.10 (0.95)	0.42 (0.92)	−0.04 (1.12)
DC2	−0.46 (0.37)	−0.25 (0.38)	−0.15 (0.49)	0.36 (0.56)	2.84 (1.54)
DC3	0.14 (0.78)	0.34 (0.77)	0.05 (0.94)	−0.90 (1.21)	−0.29 (1.26)
DC4	0.53 (0.51)	0.45 (0.47)	−1.55 (0.39)	0.44 (0.69)	−0.38 (1.10)
MVPA quintiles	3.06 (1.36)	2.82 (1.43) ^bc^	2.96 (1.39) ^bc^	3.11 (1.40) ^bc^	3.03 (1.43)

^a^. significantly different from the prudent-moderate diet group in unadjusted analyses; ^b^. significantly different from the prudent-moderate diet group in unadjusted analyses for men; ^c^. significantly different from the prudent-moderate diet group in unadjusted analyses for women. ^d^. 1000 euros, adjusted for household size. Abbreviations: BMI = body mass index; IMD = index of multiple deprivation; DC = dietary component; MVPA = moderate-to-vigorous physical activity.DC1 = prudent diet; DC2 = no wheat/dairy/eggs; DC3 = meat; DC4 = full-cream dairy.

**Table 3 nutrients-13-01607-t003:** Regression model outcomes.

		Single-Predictor Models				Multiple-Predictor Models ^a^			
		β	SE	*p*		β	SE	*p*	
Men	DC1 (prudent diet)	−0.036	0.004	<0.0001	*	−0.031	0.004	<0.0001	*
	DC2 (no wheat/dairy/eggs)	0.030	0.004	<0.0001	*	0.039	0.004	<0.0001	*
	DC3 (meat)	0.041	0.004	<0.0001	*	0.039	0.004	<0.0001	*
	DC4 (full-cream dairy)	0.023	0.004	0.0001	*	0.022	0.004	<0.0001	*
	Unhealthy vs. prudent-moderate	0.087	0.010	<0.0001	*	0.086	0.010	<0.0001	*
	Restricted vs. prudent-moderate	0.125	0.016	<0.0001	**	0.124	0.016	<0.0001	**
	Avoid meat vs. prudent-moderate	0.075	0.011	<0.0001	*	0.075	0.011	<0.0001	*
	Low-fat dairy vs. prudent-moderate	0.042	0.011	0.0002	*	0.042	0.011	0.0002	*
	MVPA quintiles	−0.007	0.004	0.0508		−0.004	0.004	0.2432	
Women	DC1 (prudent diet)	−0.043	0.003	<0.0001	*	−0.049	0.003	<0.0001	*
	DC2 (no wheat/dairy/eggs)	0.038	0.003	<0.0001	*	0.043	0.003	<0.0001	*
	DC3 (meat)	−0.017	0.003	0.0036		−0.018	0.003	0.0001	
	DC4 (full-cream dairy)	0.010	0.003	0.0023		0.011	0.003	0.0014	
	Unhealthy vs. prudent-moderate	0.097	0.009	<0.0001	*	0.095	0.009	<0.0001	*
	Restricted vs. prudent-moderate	0.156	0.013	<0.0001	**	0.156	0.013	<0.0001	**
	Avoid meat vs. prudent-moderate	0.154	0.011	<0.0001	**	0.154	0.011	<0.0001	**
	Low-fat dairy vs. prudent-moderate	0.054	0.009	<0.0001	*	0.054	0.009	<0.0001	*
	MVPA quintiles	−0.009	0.003	0.0054		−0.003	0.003	0.3343	

a. For DC1-4: adding the other dietary components and MVPA to the model; for dietary groups: adding MVPA to the model; for MVPA: adding DC1-4 to the model. ** *p* < 0.0028 & β = 0.1–0.2; * *p* < 0.0028 & β = 0.02–0.1. Abbreviations: DC = dietary component; MVPA = moderate-to-vigorous physical activity.

## Data Availability

Bona fide researchers can apply to access the UK Biobank research resource to conduct health-related research that is in the public interest. Access to the data was granted by UK Biobank following registration with their system and approval of our research project (project number 23668). All derived variables (behavioral disinhibition, four dietary components, and one dietary grouping variable) will be returned and shared through UK Biobank. For details, see https://www.ukbiobank.ac.uk/ (accessed on 7 January 2021).

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
