# Peer review of "Diet, Physical Activity, and Disinhibition in Middle-Aged and Older Adults: A UK Biobank Study"

_nutrients, 2021, doi:10.3390/nu13051607_

Round 1

Reviewer 1 Report

Thank you for your work on this interesting research topic. The research is focused on an interesting topic that could impact our understanding of disinhibited behaviour in middle aged and older adults. This cross sectional study is set out to explore how dietary patterns are related to behavioural disinhibition. The methods and results are very thorough and well explained. However, the introduction and discussion could be clearer with more literature, theoretical background and detailed evaluation and justification.

I believe that the paper in its current form needs major revisions.

Introduction

The introduction could be stronger with more background information, and a stronger rationale for why this research question is important.

  • The intro is focused around 2 hypotheses (nutrition as mediator between disinhibition and poor health, and nutrition contributing to disinhibition), but due to the cross sectional design of the study neither of these can be tested with this research study. This is acknowledged in the discussion. However, the rationale for why the current research study is important is not very clear. I think it would be more suitable to describe these as implications in the discussion, and focus the introduction more on the rationale for the current study.
  • There is a lot of previous work on eating behaviour and disinhibition that could be relevant, but is not included here. E.g. about disinhibited eating in restrained eaters, impulsivity/response inhibition related to restrained eating and unsuccessful dieting, increased food intake, unhealthy food choice, prospective weight gain.
  • Besides disinhibited eating, how is eating behaviour related to other disinhibited behaviours?
  • There is no literature included about disinhibition in middle-aged and older adults. Are they expected to be different or similar to younger adults? Eating behaviour and dietary patterns change with age, how is this expected to relate to disinhibition?
  • There is no mention of eating disorders or binge eating. For the hypothesis that nutrition may mediate the association between disinhibition and poor health, and relation to obesity (Line 43), I would expect that disinhibited eating is an important disinhibited behaviour for this research.

Materials and methods

The methods could be stronger with more justification of the chosen methods.

  • Is it possible to provide more information about the FFQ? E.g. is it validated? It is very short, which is easier for the participants, but provides limited information compared to FFQs that provide full dietary assessment including total energy intake and nutrient intakes.
  • More justification on why both PCA and latent profile analysis were performed, and why is it important that all were included in the analyses. What does it mean when the dietary groups or components are related to disinhibition? Explaining the difference would make the results easier to interpret.
  • Behavioural disinhibition does not seem to include disinhibited eating. Is there a reason this is not included?
  • Demographic characteristics included as covariates are not described as measures. They are only mentioned in the procedure.
  • Is it possible to add Cronbach’s alpha for the newly developed components to check internal reliability?

Discussion

The discussion could be stronger with more evaluation comparing results to relevant literature.

  • How do the results relate to previous work? Introduction mentions there has been work in younger adults, mainly males. Are these results similar or different from results in younger adults?
  • Anything specific about middle aged/ older adults that could explain these results, e.g. maybe food choice is related to specific health problems, or physical abilities. Physical activity levels may be low because they are restricted due to frailty, sarcopenia or other health reasons.
  • Theoretical background related to how restraint may lead to disinhibited eating, and how restraint is related to impulsivity, could be more detailed. Restraint is not always only about intention to limit food intake, it may include current dieters and is often related to a history of attempted (unsuccessful) diets.
  • References are needed for every statement throughout manuscript, e.g.:
    • Line 264-266
    • Line 279-281
    • Line 317-319
    • Line 328 – reference for potential influence genetic bias
    • Line 328-330 – reference for potential systematic bias
  • A limitation of the study is that the dietary assessment data is very limited, and there is no data on energy or nutrient intake.

Author Response

Reviewer 1

We thank the reviewer for his/her kind words and thorough review of the manuscript. The reviewer indicates several suggestions for improvement, mostly focusing on the introduction and discussion sections. We have numbered the reviewer’s suggestions and will address them one-by-one.

Introduction

  1. The intro is focused around 2 hypotheses (nutrition as mediator between disinhibition and poor health, and nutrition contributing to disinhibition), but due to the cross sectional design of the study neither of these can be tested with this research study. This is acknowledged in the discussion. However, the rationale for why the current research study is important is not very clear. I think it would be more suitable to describe these as implications in the discussion, and focus the introduction more on the rationale for the current study.

As explained in the discussion section, we fully agree with the reviewer that the hypotheses appearing early on in the manuscript cannot be tested provided the cross-sectional nature of our study. The hypotheses are in fact mentioned mainly to introduce both the complexity of relationships between diet and disinhibition (e.g. bi-directionality) and the relevance of these relationships (e.g. for physical health). We acknowledge that our original wording might have misguided the reader as to the hypotheses tested in the current study. To address this, we made the following changes:

Line 41-44: Poor self-regulation and psychiatric disorders related to disinhibition have been associated with a range of negative long-term health outcomes such as obesity [2] and cardiovascular disease [3], emphasizing the relevance of the concept.

Line 45-48: Several lines of research indicate that nutrition may be an important factor in the association between (extreme manifestations of) disinhibition and poor health. Impulsivity and poor self-regulation have been associated with unhealthy food choices, both in experimental settings [4], [5] and in observational studies [6].

In addition, we added the following sentences to the final paragraph of the introduction to clarify the aims and scope of our study:

Line 65-72: In the current study, we investigated associations between disinhibition and dietary behaviors in the unprecedentedly large UK Biobank sample. The observational and cross-sectional nature of our study does not allow causal conclusions regarding the relationship between diet and disinhibition, nor can we test mediation effects (e.g. from disinhibition to diet to poor health). By addressing several important limitations of the existing epidemiological evidence, however, we affirm and broaden the knowledge base of the potentially causal bidirectional relationships between diet and disinhibition. First, prior studies have primarily focused on overall diet quality;

  1. There is a lot of previous work on eating behavior and disinhibition that could be relevant, but is not included here. E.g. about disinhibited eating in restrained eaters, impulsivity/response inhibition related to restrained eating and unsuccessful dieting, increased food intake, unhealthy food choice, prospective weight gain. Besides disinhibited eating, how is eating behaviour related to other disinhibited behaviours?

Designing this study, we intentionally and explicitly excluded disinhibited eating from our outcome measure/phenotype to avoid spurious correlations. For details regarding this decision, please see our response to question 7 below. Disinhibited eating as such is not a phenotype of our interest. Of course, disinhibited eating may - at least in part - explain our findings, especially the findings regarding prudent diet. We now mention the potential role of disinhibited eating in the introduction and discussion sections as follows:

Line 45-54: Several lines of research indicate that nutrition may be an important factor in the association between (extreme manifestations of) disinhibition and poor health. […] Disinhibited eating, or the tendency to overeat in response to different stimuli including external food cues and stress, is associated with higher energy intake, higher BMI and elevated obesity risk [10], [11] as well as with more general indicators of behavioral disinhibition such as ADHD symptoms [12], [13] and poor executive functioning task performance [14], [15].

Line 269-313: Our finding of a negative association between prudent diet and disinhibition among middle-aged and older men and women corroborates and extends the existing disinhibition literature that tends to focus on the young, especially young males. […] The negative association between prudent diet and behavioral disinhibition might indicate an effect of disinhibition on diet quality (e.g. through disinhibited eating), an effect of diet quality on disinhibited behaviors (e.g. through cortico-limbic changes ), or both, or neither (e.g. when the association results from confounding). We also add to the available knowledge by […] in the below discussion.

  1. There is no literature included about disinhibition in middle-aged and older adults. Are they expected to be different or similar to younger adults? Eating behavior and dietary patterns change with age, how is this expected to relate to disinhibition?

Unfortunately, very little is known about manifestations of behavioral disinhibition at later age. In fact, we were unable to identify any convincing evidence showing that manifestations of behavioral disinhibition do or do not change with increasing age. As a result, we could not formulate specific hypotheses and cautiously worded our expectation as follows:

Line 75-77: Second, behavioral disinhibition is often studied in the young, especially in boys and young men. Dietary patterns differ by sex and change with age [21], [22], as may manifestations of disinhibition.

If the reviewer is aware of any relevant literature that may help address this knowledge gap, we would be very interested to hear about it and include it in the paper.

  1. There is no mention of eating disorders or binge eating. For the hypothesis that nutrition may mediate the association between disinhibition and poor health, and relation to obesity (Line 43), I would expect that disinhibited eating is an important disinhibited behavior for this research.

Please see our reply to question 2 and 7. Designing this study, we intentionally and explicitly excluded eating disorders from our outcome measure/phenotype to avoid spurious correlations.

Materials and methods

We very much appreciate the reviewer’s scrutiny reading the methods section. The reviewer identified several areas where he/she missed methodological details. Mostly, these details had been left out of the previous version of the manuscript in the interest of brevity. We have now added the requested details:

  1. Is it possible to provide more information about the FFQ? E.g. is it validated? It is very short, which is easier for the participants, but provides limited information compared to FFQs that provide full dietary assessment including total energy intake and nutrient intakes.

We have added the following sentence and reference for further information:

Line 101-104: Dietary habits were assessed using a 29-item food frequency questionnaire (FFQ) about participants’ past-year average diets. While this relatively short FFQ does not allow calculation of total energy or (macro-)nutrient intake, it has been shown to reliably estimate intakes of the main food groups [31].

  1. More justification on why both PCA and latent profile analysis were performed, and why is it important that all were included in the analyses. Explaining the difference would make the results easier to interpret.

We have added the following information:

Line 105-112: We thus performed two established data-reduction techniques: principal component analysis (PCA) to detect shared variance between items (i.e. dietary components), and latent profile analysis (LPA) to detect clusters of subjects behaving similarly with respect to the dietary components (i.e. dietary groups). Both techniques are complementary: while PCA extracts common features of diet across participants (e.g. “overall diet quality”), LPA extracts homogenous participant groups best described by (combinations of) these features. For details, see Supplementary Information (SI) section 1-2.

  1. Behavioral disinhibition does not seem to include disinhibited eating. Is there a reason this is not included?

Correct: our statistical conceptualization of behavioral disinhibition does not include disinhibited eating. As noted by the reviewer in an earlier comment, the statistical concept similarly does not include disordered eating or the presence of eating disorders. We agree with the reviewer that both disinhibited and disordered eating, as well as related concepts such as binge-eating, are in fact integral parts of the disinhibited phenotype. Statistically, however, incorporating disinhibited/disordered eating in the disinhibition phenotype would be incorrect: dietary behaviors are captured extensively in the predictor variables (dietary components and dietary groups) and therefore cannot also be part of the outcome variable (behavioral disinhibition). Including features of diet on both sides of the equation would result in spuriously high correlations. For the same reason, items regarding alcohol consumption were removed from the dietary data when computing dietary components (see line 116-117). After all, all participants scoring high on excessive drinking (part of the disinhibition construct) would also score high on alcohol intake resulting in spurious correlations between predictor and outcome.

  1. Demographic characteristics included as covariates are not described as measures. They are only mentioned in the procedure.

We have now included extra details regarding the measurement of covariates in a separate paragraph. In the interest of space, and because details regarding all variables obtained in the UK Biobank study are publicly available at http://www.ukbiobank.ac.uk, we have attempted to keep it short.

Line 171-184: 2.5. Covariates

Participants self-reported their ethnicity as white, mixed, Asian, Black, Chinese or  other, which was recoded to white/non-white. Body mass index (BMI) in kg/m2 was measured on site. Participants self-reported their habitual sleep duration in hours and minutes. As both long and short sleep duration are believed to be detrimental, sleep duration was categorized in three non-ordinal categories. Short: <7 hours/night; normal: 7-9 hours/night or 7-8 hours/night for adults age ≥65; long: >9 hours/night or >8 hours/night for adults age ≥65. Neighborhood-based index of multiple deprivation (IMD) by region (England, Scotland, Wales), an index of relative poverty of small areas based on e.g. un-employment and crime rates, was determined based on postcode of residence. Participants also self-reported their annual household income, which was adjusted for household size using equivalence factors, their unemployment status, and their number of years of education [34] (no qualifications: 7y; CSEs: 10y; O-levels/GCSEs: 10y; A levels/AS levels: 13y; other professional qualification: 15y; NVQ or HNC: 19y; college or university degree: 20y).

  1. Is it possible to add Cronbach’s alpha for the newly developed components to check internal reliability?

Cronbach’s alpha measures internal consistency, i.e. how closely related a set of items are as a group. The four newly developed components are principal component scores, i.e. all four component are based on the same set of all 58 dietary items. We could thus compute a single Cronbach’s alpha across these 58 items (Cronbach’s alpha = 0.027 [95% CI: 0.020 – 0.034], but this value reflects the internal consistency of the entire FFQ rather than the newly developed components. The fact that its value is close to zero is not surprising, provided that the FFQ intends to measure a wide range of dietary behaviors with variable associations between each other.

To provide the reviewer with an indication of how well the 58 individual items map into the four dietary components, we’re providing a table containing the uniqueness (the variance that is unique to the item i.e. not shared with other items), communality (one minus uniqueness) and complexity (number of components needed to account for the observed scores) of each item. This table has now been added as supplemental table S1.4. Together with the table of factor loadings (table S1.2) and the correlations between dietary components (table S1.3), we hope these values provide the reviewer and reader with abundant information.

Discussion

  1. How do the results relate to previous work? Introduction mentions there has been work in younger adults, mainly males. Are these results similar or different from results in younger adults?

We have added the following clarification:

Line 296-304: Our finding of a negative association between prudent diet and disinhibition among middle-aged and older men and women corroborates and extends the existing disinhibition literature that tends to focus on the young, especially young males. In line with findings among younger populations [6], we show that behavioral disinhibition is associated with unhealthier food choices among middle-aged and older individuals. The negative association between […] results from confounding).

  1. Anything specific about middle aged/ older adults that could explain these results, e.g. maybe food choice is related to specific health problems, or physical abilities. Physical activity levels may be low because they are restricted due to frailty, sarcopenia or other health reasons.

We thank the reviewer for this question. The question is especially relevant to two of our findings: the general weakness of identified effects, and the absence of associations between behavioral disinhibition and moderate-to-vigorous physical activity. In these two instances, we added the following sentences:

Line 304-313: We also add to the available knowledge by qualifying the association between prudent di-et and disinhibition as ‘very weak’. […] The weak association strength of this and all other associations detected in our study might be related to the relatively old age of participants. Other factors specifically relevant in older participants (e.g. frailty or health problems) might explain a relatively large proportion of variance in dietary behaviors, leaving less variability to be explained by behavioral disinhibition. The weak association strength of all findings should be kept in mind when reading the below discussion.

Line 355-365: Finally, the absence of an association between MVPA and disinhibition is of note. […] The absence of associations with MVPA might be related to the relatively old age of the current sample, as health problems and frailty might restrict their activities. Thorough investigation of specific physical activity characteristics is beyond the scope of the current paper, and is recommended for future studies.

  1. Theoretical background related to how restraint may lead to disinhibited eating, and how restraint is related to impulsivity, could be more detailed. Restraint is not always only about intention to limit food intake, it may include current dieters and is often related to a history of attempted (unsuccessful) diets.

We agree with the reviewer that the description of dietary restraint benefits from more background information. We have now added the following information:

Line 327-336: Studies of dieting/weight control provide yet another explanation: highly-impulsive individuals tend to score higher on dietary restraint. Restraint is a cognitive concept characterized by the (successful or unsuccessful) intention to limit food intake, that has been associated with dieting, past attempts at dieting, and weight fluctuations [39]. Although participants were instructed to report dietary habits, not intentions, it is plausible that dieters might report the dietary pattern they (successfully or unsuccessfully) intend to adhere to. Finally, higher endorsement […] the never-criterion.

  1. References are needed for every statement throughout manuscript.

The reviewer suggests five instances in which a reference would be needed. The first three instances, however, describe our current findings, putative interpretations thereof, or recommendations for future studies. As these statements do not refer to prior work (of others), we are unable to provide references for the following statements:

Line 264-266 (now line 294-295): All associations between diet and disinhibition were very weak (β<0.2). Physical activity was not associated with disinhibition.

Line 279-281 (now line 318-321): Note that the restricted diet group was also characterized by lower overall diet quality compared to the prudent-moderate diet group, potentially suggesting a more general lack of nutrients or residual confounding.

Line 318-319 (now line 364-365): Thorough investigation of specific physical activity characteristics is beyond the scope of the current paper, and is recommended for future studies.

The final two instances pointed out by the reviewer did contain sentences that were suitable for referencing. We have added the following references:

Line 328 (now line 377-380): Another inherent vulnerability of observational studies is the possibility of unmeasured confounding, i.e. both disinhibition and dietary behaviors might be driven by an unmeasured third variable (e.g. genetic confounding [51]).

Line 328-330 (now line 383-385): Finally, dietary habits and physical activity levels were self-reported and might be subject to systematic bias [52] (e.g. less accurate reporting in high-impulsive participants).

We suspect that automated (re)numbering of the lines might have altered the numbers in either our manuscript version or that of the reviewer. We therefore scrutinized the discussion section looking for more statements that might require additional references. We added the following references:

Line 324-325: Alternatively, elimination of specific foods may suggest a higher prevalence of (allergic) food intolerances among disinhibited or impulsive individuals [19].

Line 351-352: The early-malnutrition [37] and food-intolerance hypotheses [20] may thus apply to the meat-avoiding group as well.

Line 352-354: Alternatively, the decision to (partially) eliminate meat from one’s diet might be influenced by fluctuating societal trends, to which impulsive individuals may be more sensitive [46].

  1. A limitation of the study is that the dietary assessment data is very limited, and there is no data on energy or nutrient intake.

We agree with the reviewer. This limitation is now mentioned in the methods section (see question 5) as well as in the discussion section:

Line 373-383: The current study has limitations as well. […] We were unable to adjust analyses for total energy intake, as the short FFQ does not allow calculation of this metric. Moreover, no data […] brain development.

Reviewer 2 Report

The purpose of the study was to investigate the association between dietary patterns and exercise with disinhibition. The manuscript is very well written, and the topic is very interesting. Here are my suggestions to improve the manuscript.

  1. If the authors are able to include a concise paragraph on the neuroscience of disinhibition in the introduction, it will help make sense of how the diet is related to neurobehaviors.
  2. please provide more information on the criteria for the component’s selection (scree plot, eigenvalue,etc..).
  3. The findings on meat is interesting, which suggests that meat impacts neurobehaviors in absence or presence of certain nutrients, which may affect levels of neurotransmitter related to human behavior such as serotonin and dopamine.
  4. As for exercise, VPA is a source of cortisol, which typically increases impulsivity.
  5. It would be also helpful to tie some of the food groups  to neurotransmission to explain disinhibition, if this topic fits within the authors’ domain. The discussion is good, but it will be richer with such information.
  6. One major limitation to add is the study did not account for the morphological structures of brain regions that control impulsivity and similar behavior. The fact that the studied cohort is middle-aged, there is a potential discrepancy in brain volume as drinking, long-term stress, poor quality diet (nutritional deficiencies, inflammation and oxidative stress), can impact brain volume differently; hence functionality. 
  7. Another limitation is not knowing the nutritional history of the cohort (per studied patterns) that may have impacted the normal development of the brain and which may result in higher disinhibition. 

Author Response

Reviewer 2

We thank the reviewer for his/her careful reading of our manuscript. Below we address the reviewer’s suggestions for improvement one-by-one. All changes to the original manuscript have been underlined.

  1. If the authors are able to include a concise paragraph on the neuroscience of disinhibition in the introduction, it will help make sense of how the diet is related to neurobehaviors.

We thank the reviewer for this suggestion. Although neurobiology is not a main focus of the current paper, we agree that a brief introduction of the most relevant pathway would be beneficial. We have added the following sentences to the introduction:

Line 55-64: While a causal pathway running from disinhibition/self-regulation to dietary habits is plausible, nutrition may also contribute to disinhibited behaviors. […] Excessive consumption of palatable foods may evoke lasting changes in cortico-limbic dopamine signaling that are reminiscent of patterns associated with addiction [18]. Dopaminergic cortico-limbic systems are implicated not only in addiction, but also in emotion regulation, reward processing, and impulse control. Others have speculated […] specific foods/allergens [20].

  1. Please provide more information on the criteria for the component’s selection (scree plot, eigenvalue, etc..).

Selection of the dietary components was performed manually, starting with a single component model and adding components one-by-one. Components were retained as long as they remained unique, interpretable and stable. We agree with the reviewer that the current description of component selection is rather brief and would benefit from more details. We have therefore added the following sentences:

Line 119-126: The resulting 58 food items/contrasts, regressed on age and sex, were entered in a PCA with Promax rotation on the residuals. Component selection was performed manually (LJSS), starting with a single component and adding components one-by-one. Components were retained as long as they were unique (no prior component reflected the same set of items), interpretable (items loading on the component form plausible groupings) and stable (the component remained upon adding the next component). Component selection was verified with a second author (CAH).

  1. The findings on meat is interesting, which suggests that meat impacts neurobehaviors in absence or presence of certain nutrients, which may affect levels of neurotransmitter related to human behavior such as serotonin and dopamine.

We have added the reviewer’s suggestion to the discussion section as follows:

Line 337-348: Our findings regarding meat consumption were contradictory. […] As meat is rich in tryptophan, a precursor of dopamine, meat consumption might affect neurotransmitter levels in brain systems important for inhibitory control. Higher disinhibition in the […] were similar (data not shown).

  1. As for exercise, VPA is a source of cortisol, which typically increases impulsivity.

The reviewer’s comment is interesting: he/she suggests a positive relationship between vigorous physical activity (VPA) and impulsivity via the cortisol response to exercise. This hypothesis does not match with our findings:  we found a (very weak) negative association between VPA and behavioral disinhibition, i.e. those who performed more exercise were less disinhibited. This discrepancy can possibly be explained by the duration and timing of cortisol effects: the acute increased cortisol response to VPA is short-lived, peaking at approximately 20-30 minutes after exercise cessation, and is followed by longer-lasting (24-48h) depression of cortisol levels. Baseline cortisol levels may thus be lower in those who perform VPA regularly. Importantly, however, we also found that the relationship between VPA and disinhibition disappeared when adding the dietary components to the model, suggesting that the association was better accounted for by dietary behaviors. We have included the following sentence:

Line 355-360: In line with […] we found a very weak association between disinhibition and VPA but not MPA. Regular exercise might lower baseline cortisol levels [48], which in turn may be associated with lower impulsivity [49]. However, the association between VPA and disinhibition was partially accounted for by dietary factors.

  1. It would be also helpful to tie some of the food groups to neurotransmission to explain disinhibition, if this topic fits within the authors’ domain. The discussion is good, but it will be richer with such information.

We thank the reviewer for this interesting suggestion. Unfortunately, the effects of specific food groups on neurotransmission are not within the scope of our expertise. Where possible, we tentatively mention the potential effects of food groups on neurotransmitters, hormones and/or brain function throughout the discussion (e.g. line 303, 343, 358). A more elaborate discussion would require specific expertise.

  1. One major limitation to add is the study did not account for the morphological structures of brain regions that control impulsivity and similar behavior. The fact that the studied cohort is middle-aged, there is a potential discrepancy in brain volume as drinking, long-term stress, poor quality diet (nutritional deficiencies, inflammation and oxidative stress), can impact brain volume differently; hence functionality.

We fully agree with the reviewer that the absence of brain correlates at this point is a limitation of our paper. In fact, we are currently collaborating with experts in the field of neuroimaging to integrate the UK Biobank structural neuroimaging data into our investigation. The first results of this endeavor will hopefully emerge soon. We have added the limitation as follows:

Line 373-377: The current study has limitations as well. The cross-sectional and observational nature of our study precludes causal inferences. Observed associations might reflect an effect of disinhibition on dietary choices, an effect of dietary intake on disinhibited behaviors, or both. In future studies, integrating structural brain imaging data could be an important asset to elucidate the mechanisms underlying the observed associations.

  1. Another limitation is not knowing the nutritional history of the cohort (per studied patterns) that may have impacted the normal development of the brain and which may result in higher disinhibition.

We agree with the reviewer that this is a limitation of our study. Unfortunately, UK Biobank provides no data regarding nutritional history or early life exposure to malnutrition. We have added the following sentence:

Line 373-383: The current study has limitations as well. […] We were unable to adjust analyses for total energy intake, as the short FFQ does not allow calculation of this metric. Moreover, no data is available regarding nutritional history or exposure to early life malnutrition, both of which may have impacted on normal brain development.

Reviewer 3 Report

Dear Authors,

The manuscript is interesting and well written. It needs  some minor changes.

The introduction is well written and include all relevant references.

In line 133 please specify what the authors mean about moderate and vigorous physical activity.

Please check the references.

Author Response

Reviewer 3

We thank the reviewer for his/her compliments regarding the manuscript and especially the introduction. The reviewer indicates two minor suggestions for improvement.

  1. In line 133 please specify what the authors mean about moderate and vigorous physical activity.

We have added the following clarification to the description of MVPA (underlined):

Line 154-165: Participants self-reported their frequency (days/week) and duration (minutes/day) of engaging in moderate and vigorous physical activity. In the questionnaire, moderate physical activity (MPA) was described as “activities like carrying light loads, cycling at normal pace (do not include walking) for 10 minutes or more.” Vigorous physical activity (VPA) was described as “activities that make you sweat or breathe hard such as fast cy-cling, aerobics, heavy lifting for 10 minutes or more.” MVPA was calculated as the sum of moderate and vigorous physical activity in minutes/week.

  1. Please check the references.

We have checked all references again and corrected any errors we encountered.